# Antioxidant Activity and Peptide Levels of Water-Soluble Extracts of Feta, Metsovone and Related Cheeses

Athina Kalle, Ioannis Lambropoulos, Konstantinos Bourazas [†] and Ioannis G. Roussis *[ID]

Laboratory of Food Chemistry, Department of Chemistry, University of Ioannina, 45110 Ioannina, Greece; kalleathina@gmail.com (A.K.); glambrop@yahoo.gr (I.L.); bourazaskostas@gmail.com or bourazas.konstantinos@ucy.ac.cy (K.B.)
* Correspondence: iroussis@uoi.gr; Tel.: +30-2651008344
† Current address: Department of Mathematics and Statistics, KIOS Research and Innovation Center of Excellence, University of Cyprus, Nicosia 2109, Cyprus.

**Featured Application: The investigation of antioxidant activity of cheeses has unveiled an array of promising applications in the food industry and promotion of consumer health. Cheese, as a widely consumed dairy product, possesses antioxidants that not only contribute to its preservation, but also enrich the quality and health benefits for the consumers.**

**Abstract:** The purpose of the present study was to evaluate the antioxidant, anti-inflammatory activity and peptide levels of Feta cheese and other brined cheeses, and Metsovone cheese and other smoked cheeses. Feta, goat and cow cheeses are classified as brined. Feta cheese is made exclusively from ewe milk or ewe milk with the addition of a small amount of goat milk. Metsovone and other smoked cheeses are made from cow milk with a possible addition of small amounts of ewe and goat milk. The antioxidant activity was determined using Folin and FRAP assays, while the peptide content was determined using Bradford and Lowry assays. The anti-inflammatory activity was assessed using the lipoxygenase inhibition assay. The assays were applied in the water-soluble extract of cheeses. The results showed that Feta cheese and brined cow cheese differed in antioxidant activity. Feta cheese and brined goat cheese also differed in both antioxidant activity and peptide levels. Specifically, Feta cheese had higher antioxidant activity in comparison with both goat and cow cheeses. As for peptide content, Feta cheese had a higher peptide level compared to goat cheese. The results also showed that Metsovone cheese and other smoked cheeses exhibited significant antioxidant activity and peptide levels. Moreover, the water-soluble extracts of all cheeses showed some anti-inflammatory activity.

**Keywords:** Feta cheese; Metsovone cheese; antioxidant activity; peptide content; anti-inflammatory activity

## 1. Introduction

Cheese is a fermented milk product that is produced globally in different varieties. The raw material, milk, used in cheese production is mainly cow, goat and sheep milk. The characteristics of cheeses are dependent on the composition and quality of the milk, as well as various other factors [1,2]. Cheese has gained special interest due to its positive effects on health. In some recent studies, health beneficial properties, such as antioxidant and anti-hypertensive activities, have been proven [3–5].

Feta cheese is recognized as a Protected Designation of Origin (PDO) product and belongs to brined cheeses [6]. It is made exclusively from ewe milk or a mixture of ewe and up to 30% goat milk. Metsovone is a pasta filata type of cheese, produced from cow milk or a mixture of cow and up to 20% goat and ewe milk [7]. Metsovone cheese is classified as smoked cheese [8].

The water-soluble extract (WSE) of cheese contains mainly small and medium-sized peptides, free amino acids and salts. The water is an efficient solvent for extracting small

peptides from cheese, many of which play a crucial role in bioactivities. The level of water-soluble nitrogen is used for the cheese ripening index, as it tends to increase during the ripening process [9,10].

During the ripening process, biochemical and chemical reactions occur, which contribute to the formation of the characteristic texture and flavor of each cheese. The main milk proteins, αs1-casein and β-casein are degraded into larger and smaller peptides, that influence the texture and flavor of cheeses, through biochemical factors. The larger peptides and oligopeptides enhance cheese flavor, as they are substrates for the enzymatic activities of the microbiota. Peptides are components that are generated and degraded during cheese production, ripening, and preservation [11–14].

Cheese contains antioxidants that inhibit oxidative processes in the cheese and neutralize free radicals, prevent oxidative changes and support the body's defense mechanisms. It has been reported that amino acids in dairy products are able to act as antioxidants and especially hydrogen donors [15]. Peptides, which have positive effects on metabolic functions and provide health benefits, are defined as bioactive peptides. These peptides have various bioactivities, such as antioxidant, antimicrobial, anti-inflammatory, and other actions [16–18]. Gupta et al. [19] showed an increase of antioxidant activity of Cheddar cheese up to the fifth month and then it decreased, suggesting that antioxidant peptides were not resistant to further proteolysis. The antioxidant activity was determined through different assays such as 2,2′-azinobis (3 ethyl-benzothiazoline)-6-sulphonic acid (ABTS), 2,2-diphenyl-picrylhydrazyl (DPPH) and superoxide radical scavenging activity. Bottesini et al. [20] investigated the antioxidant activity of Parmigiano Reggiano cheese, which is mostly due to antioxidant-free amino acids and to a minor extent, antioxidant peptides. Hernández-Ledesma et al. [21] identified that peptides contained at least one proline residue, suggesting that these peptides were likely responsible for the antioxidant activity that was found in fermented milk. In these studies, the antioxidant activity was determined by ABTS and chromatographic assays.

Antioxidants are also commonly used to inhibit the action of lipoxygenase and prevent autoxidation of substrates. Lipoxygenases exist in the human body and play a significant role in inflammation. Therefore, the inhibition of lipoxygenases is considered important for disease prevention. Inflammation is associated with diseases, such as cancer, diabetes, asthma, atherosclerosis, and neurodegenerative diseases [22]. Rival et al. [23] reported peptides, derived from bovine casein, responsible for inhibition of lipoxygenase activity.

The purpose of this study was to evaluate the antioxidant activity and peptide profile of Feta cheese and other brined cheeses, as well as Metsovone cheese and other smoked cheeses. In addition, the anti-inflammatory activity was evaluated. The antioxidant activity was determined using Folin and FRAP assays, while the peptide levels were determined using Bradford and Lowry assays. The anti-inflammatory activity was determined using the lipoxygenase inhibition assay.

## 2. Materials and Methods

### 2.1. Cheese Samples

In this study, triplicate cheese samples were sourced from a selection of diverse dairy industries located across Greece. Feta samples were obtained from Dodoni (Epirus1), Karalis (Epirus2), Bizios (Elassona, Thessaly) and Erymanthos (Achaia, Western Greece). Goat cheeses were collected from Dodoni (Epirus1), Karalis (Epirus2), Exarchos (Elassona, Thessaly) and Erymanthos (Achaia, Western Greece). Cow cheeses were supplied from Dodoni (Epirus1), Bizios (Elassona, Thessaly) and Vlacha (Central Macedonia). Metsovone samples were from Tositsa Foundation (Epirus) and the other smoked cheeses were supplied from Pappas (Epirus) and BelasVermion (Central Macedonia).

### 2.2. Reagents

Folin–Ciocalteu's phenol reagent 2N was purchased from Sigma-Chemicals (St. Louis, MO, USA). Gallic acid, bovine serum albumin (BSA) 96% and sodium carbonate (anhy-

drous) were purchased from Merck (Darmstadt, Germany). The reagent 2,4,6-tri(2-pyridyl)-1,3,5-triazine (TPTZ) was from Fluka (Buchs, Switzerland). Hydrochloric acid $\geq$ 37%, lipoxygenase and linoleic acid were from Sigma–Aldrich (Steinheim, Germany). Acetic acid was purchased from Carlo Erba (Val de Reuil, France). Boric acid and L(+)-ascorbic acid (Vitamin C) 99.7% were from Riedel-de Haën (Seelze, Germany), while the Bradford reagent was purchased from Roti (Karlsruhe, Germany). Ethanol $\geq$ 98% was purchased by Honeywell (Charlotte, NC, USA). HPLC-grade water and acetonitrile were purchased from Chem-Lab (Zedelgem, Belgium).

### 2.3. Instruments and Apparatus

The absorbance was measured using a three-decimal precision spectrophotometer, Jenway 6505 UV/VIS (Stone, Staffordshire, UK). The pH measurements were obtained using a Consort C831 pH-meter (Turnhout, Belgium), with a precision of two decimal points. The weight of materials was measured using an electronic balance, Kern ABS balance (Lohmar, Germany), with a precision of four decimal points. The Hettich Zentrifugen Mikro 22R centrifuge (Tuttlingen, Germany) and Stomacher, Bag Mixer 400 (Puycapel, France), were used for extract preparation. Waters 660E HPLC System with Diode Array Detector Waters 996 (Massachusetts, UK) was used.

### 2.4. Methods of Analysis

#### 2.4.1. Cheese Extract Preparation

Water-soluble extracts (WSEs) of cheese samples were prepared using the method of Kuchroo and Fox with some modifications [24,25]. The 10 g cheese sample was homogenized in 20 mL of deionized water for 5 min using the Stomacher 400 and the resulting homogenate was incubated in a 40 °C water bath for 1 h. Then, the mixture was centrifuged at 5500 rpm at 4 °C for 20 min. The supernatant was filtered through glass wool to remove residual suspended fat and then, through Whatman filter paper No. 42. Before analysis, the permeated was filtered using a 0.45 μm membrane filter. WSEs were refrigerated and analyzed within 4 h by Folin, FRAP, Bradford, Lowry and anti-inflammatory assays. The samples for HPLC analysis were preserved by freezing.

#### 2.4.2. Antioxidant Activity Assays

The antioxidant activity was determined using Folin–Ciocalteu assay [26,27]. In a test tube, 1.9 mL of deionized water, whose pH was adjusted to the pH of the WSE, was mixed with 100 μL of WSE, and followed by the addition of 125 μL of Folin–Ciocalteu reagent and vortexed. After 1 min, 380 μL of 20% ($w/v$) sodium carbonate were added, the mixture was vortexed and kept in the dark at room temperature for 2 h. The absorbance of the mixture was measured at 750 nm against the blank. The blank was prepared by adding deionized water with the same pH as the WSE's pH instead of the sample. Gallic acid (GAE) was used as a standard and the results were expressed as mg/L GAE equivalents. They were also expressed as mg GAE/kg cheese. The equation of the linear standard curve was y = 0.0041x + 0.0083, $R^2$ = 0.9992.

The antioxidant activity was determined using the FRAP assay [24,28]. The FRAP reagent was freshly prepared by mixing 10 parts of 300 mM acetate buffer (pH 3.6), 1 part of 9 mM 2,4,6-tripyridal-s-triazine (TPTZ) in 0.05 M HCl, and 1 part of 20 mM $FeCl_3$ in 0.05 M (in the ratio of 10:1:1 vol/vol/vol). The FRAP assay was conducted following the methodology described in the study by Perna et al. [20], with specific modifications. The difference was that 1.5 mL of FRAP reagent and 0.25 mL WSE were mixed, and the blank reagent was prepared by adding deionized water with the same pH as the WSE's pH instead of the sample. Ascorbic acid (VC) was used as a standard [29] and the results were expressed as mg/L VC equivalents. Moreover, they were expressed as mg VC/kg cheese. The equation of the linear standard curve was y = 0.0124x + 0.0042, $R^2$ = 0.9992. The standard curve of GAE was also constructed and the equation of the line was y = 0.1242x + 0.0017, $R^2$ = 0.9985.

### 2.4.3. Peptide Assays

The peptide content was calculated using the Bradford assay [25,30]. In an Eppendorf tube, 0.75 mL of Bradford reagent (commercial reagent 4-fold diluted with deionized water) was mixed with 0.25 mL WSE. Then, the mixture was incubated at room temperature for 5 min. After 5 min, the absorbance of the mixture was measured at 595 nm against the blank (containing 0.25 mL of deionized water with the same pH as the WSE's pH instead of the WSE). Bovine serum albumin (BSA) was used as a standard and the results were expressed as mg/L BSA equivalents. They were also expressed as mg BSA/kg cheese. The equation of the linear standard curve was $y = 0.0043x - 0.0013$, $R^2 = 0.9986$.

The peptide content was calculated using the Lowry assay [31,32]. Prior to the analysis, the reagent of Lowry was freshly prepared by mixing 50 parts of reagent A: 2% ($w/v$) $Na_2CO_3$ in 0.1 M NaOH, and 1 part of reagent B: 0.5% ($w/v$) $CuSO_4 \cdot 5H_2O$ in 1% ($w/v$) potassium tartate. In an Eppendorf tube, 200 μL deionized water, whose pH has been adjusted to the pH of the WSE, was mixed with 50 μL of WSE. Then, 1.25 mL Lowry reagent was added, and the mixture was vortexed. After 10 min incubation in the dark at room temperature, 0.25 mL of diluted (1 M) Folin–Ciocalteu reagent was added, and the mixture was kept in the dark for 30 min. After 30 min, the absorbance of the mixture was measured at 750 nm against the blank (containing 50 μL of deionized water with the same pH as the WSE's pH instead of the WSE). Bovine serum albumin was used as a standard and the results were expressed as mg/L BSA equivalent. Moreover, they were also expressed as mg BSA/kg cheese. The equation of the linear standard curve was $y = 0.0002x + 0.0314$, $R^2 = 0.9971$.

The peptide profile was evaluated through HPLC analysis [33,34]. Waters 660E HPLC system and the column, MZ Analysentechnik C18 (4.6 mm × 25.0 cm) were used. The Empower 2 Software Build 2154 (version 6.20.00.00) was used for the analysis. The elution was a gradient, using solvents A (0.1% $v/v$ trifluoroacetic acid in HPLC-grade water) and B (0.09% $v/v$ trifluoroacetic acid in a 60:40 $v/v$ mixture of acetonitrile and HPLC-grade water). Solvent A was 100% for 5 min, followed by a change in composition from 0% to 100% of solvent B for 20 min, and finally, 100% solvent B for 13 min. The flow rate was 0.8 mL/min. Before injection, the samples were diluted 10 times with deionized water. A volume of 20 μL of diluted WSE was injected into the column and the peptide elution was monitored at 214 nm.

### 2.4.4. Anti-Inflammatory Assay

The anti-inflammatory activity was determined using the lipoxygenase (LOX) inhibition assay [23,35]. In an Eppendorf tube, 900 μL of borate buffer 1 M pH = 9, 100 μL of WSE and 100 μL of LOX (diluted in borate buffer) were mixed. Then, the mixture was vortexed and kept in the dark for 5 min. After 5 min, 100 μL of 4.18 mM linoleic acid in ethanol (≥98%) was added. The absorbance was followed for 20 min at 234 nm and the absorbance at 20 min was used for the estimation of LOX activity. The control contains deionized water, whose pH was adjusted to the pH of the WSE, instead of the sample. The control of the blank contains ethanol instead of linoleic acid, and the deionized water has the same pH as the WSE instead of the sample. The anti-inflammatory activity was expressed as inhibition of lipoxygenase and was estimated through the formula:

$$\% \text{ Inhibition} = (A\text{control} - A\text{sample})/A\text{control}$$

where Acontrol and Asample represent the absorbance of the control and sample solutions, respectively.

### 2.5. Statistical Analysis

For every pairing of cheese and region, we collected n = 3 independent replicates for each measurement, whenever feasible. To assess the statistical distinctions among the types of cheese concerning various regions, we employed mixed-effects models. Specifically,

we treated the type of cheese as a fixed effect (to compare measurements across specific cheese types), while we assumed the region to be a random effect (considering the observed regions as a random sample of all possible region where these cheeses are made). To compare the pairs of different cheese types, we utilized the Kenward–Roger (K-R) Degrees of Freedom Approximation [36,37].

Statistical analysis was performed using RStudio 4.3.1. and the results of the fitted models are detailed in Section 3.

In the following Tables 1–3, we present the sample mean along with the corresponding standard deviation, while in Table 4, we present the findings of the fitted models. Specifically, we provide the pairwise expected differences between the type of cheeses for each measurement, the corresponding standard errors (in parenthesis), and the *p*-values (in italics). The reported *p*-values are adjusted so that each fitted model has a 5% type I error in total (controlling the Familywise Error Rate) so that we can be more confident about the statistical significance of the differences. Continuing with Tables 5 and 6, they provide similar information as Tables 1–3 and tabulate the sample mean and the standard deviation of the under-study variables.

**Table 1.** Antioxidant activity (Folin and FRAP assays) and peptide content (Bradford and Lowry assays) of water-soluble extracts of ripened Feta cheese from different regions of Greece.

| Region | Folin (mg GAE/L WSE) | FRAP (mgVC/L WSE) | Bradford (mg BSA/L WSE) | Lowry (mg BSA/L WSE) |
|---|---|---|---|---|
| Epirus 1 | 179 ± 7 | 37.2 ± 0.2 | 1627 ± 28 | 3777 ± 114 |
| Epirus 2 | 130 ± 3 | 35.4 ± 0.9 | 1487 ± 55 | 2994 ± 74 |
| Thessaly | 201 ± 2 | 39.7 ± 0.6 | 2468 ± 81 | 4890 ± 123 |
| Western Greece | 162 ± 4 | 34.9 ± 0.6 | 1220 ± 31 | 3204 ± 96 |

The results are presented as mean ± SD (standard deviation). Three different cheese samples (n = 3) were analyzed. The results were also expressed as mg GAE and mg VC/kg cheese for antioxidant activity and mg BSA/kg cheese for peptide content.

**Table 2.** Antioxidant activity (Folin and FRAP assays) and peptide content (Bradford and Lowry assays) of water-soluble extracts of ripened brined goat cheeses from different regions of Greece.

| Region | Folin (mg GAE/L WSE) | FRAP (mg VC/L WSE) | Bradford (mg BSA/L WSE) | Lowry (mg BSA/L WSE) |
|---|---|---|---|---|
| Epirus 1 | 110 ± 3 | 17.6 ± 0.1 | 782 ± 28 | 2200 ± 67 |
| Epirus 2 | 129 ± 3 | 27.4 ± 0.5 | 808 ± 30 | 2662 ± 75 |
| Thessaly | 118 ± 1 | 27.5 ± 0.8 | 805 ± 25 | 3007 ± 72 |
| Western Greece | 148 ± 4 | 28.5 ± 0.7 | 900 ± 35 | 2974 ± 51 |

The results are presented as mean ± SD. Three different cheese samples (n = 3) were analyzed. The results were also expressed as mg GAE and mg VC/kg cheese for antioxidant activity and mg BSA/kg cheese for peptide content.

**Table 3.** Antioxidant activity (Folin and FRAP assays) and peptide content (Bradford and Lowry assays) of water-soluble extracts of ripened brined cow cheeses from different regions of Greece.

| Region | Folin (mg GAE/L WSE) | FRAP (mg VC/L WSE) | Bradford (mg BSA/L WSE) | Lowry (mg BSA/L WSE) |
|---|---|---|---|---|
| Epirus | 100 ± 3 | 19.7 ± 0.5 | 1617 ± 34 | 2277 ± 70 |
| Thessaly | 184 ± 6 | 29.0 ± 0.7 | 2542 ± 46 | 4520 ± 65 |
| Central Macedonia | 133 ± 3 | 27.4 ± 0.6 | 1418 ± 49 | 3190 ± 81 |

The results are presented as mean ± SD. Three different cheese samples (n = 3) were analyzed. The results were also expressed as mg GAE and mg VC/kg cheese for antioxidant activity and mg BSA/kg cheese for peptide content.

**Table 4.** Expected differences among brined cheeses for antioxidant activity and peptide content.

| Cheeses | Folin (mg GAE/L WSE) | FRAP (mg VC/L WSE) | Bradford (mg BSA/L WSE) | Lowry (mg BSA/L WSE) |
|---|---|---|---|---|
| Feta–Cow | 30.42 (11.37) 0.01 | 11.76 (1.39) 0.00 | −194 (139.06) 0.17 | 546.5 (235.81) 0.06 |
| Feta–Goat | 42.08 (8.81) 0.00 | 11.60 (1.06) 0.00 | 876.67 (105.46) 0.00 | 1005.25 (178.87) 0.03 |
| Cow–Goat | 11.65 (11.37) 0.31 | −0.16 (1.39) 0.91 | 1070.66 (139.06) 0.00 | 458.75 (235.81) 0.06 |

Twelve different Feta cheese samples (n = 12), twelve different goat cheese samples (n = 12), nine different cow cheese samples (n = 9) were analyzed. The results were also expressed as mg GAE and mg VC/kg cheese for antioxidant activity and mg BSA/kg cheese for peptide content. The dark red color corresponds to a strong statistically significant difference (more than 99%), the dark blue corresponds to a moderate statistically significant difference (between 95–99%), and the yellow corresponds to a mentionable difference (between 90–95%).

**Table 5.** Antioxidant activity (Folin and FRAP assays) and peptide content (Bradford and Lowry assays) of water-soluble extracts of Feta cheese during ripening and storage.

| Age, Days | Folin (mg GAE/L WSE) | FRAP (mg VC/L WSE) | Bradford (mg BSA/L WSE) | Lowry (mg BSA/L WSE) |
|---|---|---|---|---|
| 2 | 120 ± 3 | 32.5 ± 1.5 | 2212 ± 43 | 2204 ± 62 |
| 16 | 142 ± 5 | 35.6 ± 0.7 | 1245 ± 72 | 2577 ± 63 |
| 60 | 179 ± 7 | 37.2 ± 0.2 | 1627 ± 28 | 3777 ± 114 |
| 120 | 217 ± 3 | 34.0 ± 0.7 | 1767 ± 64 | 4247 ± 113 |

The results are presented as mean ± SD. Three different cheese samples (n = 3) were analyzed. The results were also expressed as mg GAE and mg VC/kg cheese for antioxidant activity, and mg BSA/kg cheese for peptide content.

**Table 6.** Antioxidant activity (Folin and FRAP assays) and peptide content (Bradford and Lowry assays) of water-soluble extracts of Metsovone cheese, 3 and 6 months, and other ripened smoked cheeses, 3 months, from different regions of Greece.

| Region | Folin (mg GAE/L WSE) | FRAP (mg VC/L WSE) | Bradford (mg BSA/L WSE) | Lowry (mg BSA/L WSE) |
|---|---|---|---|---|
| Metsovone, 3 months | 291 ± 12 | 74.8 ± 1.9 | 3277 ± 99 | 5850 ± 153 |
| Metsovone, 6 months | 437 ± 5 | 95.8 ± 2.7 | 3478 ± 70 | 7664 ± 103 |
| Epirus, 3 months | 230 ± 8 | 51.3 ± 2.1 | 2550 ± 114 | 5110 ± 151 |
| Central Macedonia, 3 months | 294 ± 5 | 94.4 ± 0.8 | 2872 ± 111 | 6677 ± 96 |

The results are presented as mean ± SD. Three different cheese samples (n = 3) were analyzed. The results were also expressed as mg GAE and mg VC/kg cheese for antioxidant activity and mg BSA/kg cheese for peptide content.

The key finding indicates a statistically significant difference for most studied variables between Feta cheese and the others (cow and goat). Results are color-coded according to statistical significance: dark red indicates a highly statistically significant difference (greater than 99%), dark blue indicates a moderately statistically significant difference (between 95% and 99%), and yellow indicates a mentionable difference (between 90% and 95%).

## 3. Results and Discussion

### 3.1. Feta Cheese and Other Ripened Brined Cheeses

In Tables 1–3, the antioxidant activity and peptide levels of water-soluble extracts of Feta cheese, brined goat cheeses, and brined cow cheeses from different regions of Greece are presented.

The water-soluble extracts of brined cheeses exhibited antioxidant activity. The values of the peptide content determined by the Lowry assay was observed to be higher in comparison to the values of the peptide content determined by the Bradford assay. The difference of values between the Bradford and Lowry assays may be attributed to the differences in their sensitivity. Specifically, the lower limit of detection for the Bradford assay is approximately 3 to 5 kDa and therefore, small peptides and amino acids do not bind. On the other hand, the method of the Lowry assay is sensitive to the presence of amino acids and peptides [38]. Perna et al. [24] reported antioxidant activity in WSEs of cow cheeses.

In Table 4, statistical analysis of the results for antioxidant activity and peptide content of Feta cheese, brined goat cheese, and brined cow cheese, is presented.

Feta cheese and cow cheese showed statistically significant differences in antioxidant activity, determined by both Folin and FRAP assays. The antioxidant activity of Feta cheese was higher than brined cow cheese. Feta cheese and goat cheese differed significantly in antioxidant activity, determined by Folin and FRAP assays, and peptide content, determined by Bradford and Lowry assays. The antioxidant activity and peptide content of Feta cheese were higher than that of brined goat cheeses. The statistical analysis for the results expressed per kg of cheese, showed the same statistical differences with the results, expressed per L of WSE, with the following exception. In the Lowry assay, cow cheeses exhibited statistically higher values of BSA per kg of cheese than goat ones, while their values of BSA per L of WSE did not exhibit statistically significant differences.

The antioxidant activity of cow, goat and ewe milk is investigated with various assays, including FRAP by Stobiecka et al. [18]. The research showed differences in antioxidant activity between different types of milk due to their chemical composition differences. Goat milk had the highest antioxidant activity and cow milk had the lowest antioxidant activity. Perna et al. [24] determined the antioxidant activity of cow cheeses using the FRAP assay, which was increased during the ripening process. Gupta et al. [19] showed that significant peptide levels of WSEs of Cheddar cheese were increased throughout the ripening process.

In Figure 1, the typical chromatograms of Feta, goat and cow cheese are presented.

The chromatograms of all brined cheeses exhibited two similar peak regions. In the first region, two high peaks appeared, corresponding to hydrophilic peptides, at similar retention times. In the second region, which corresponds to more hydrophobic peptides, a cluster of different peaks was observed within a similar retention time range. However, there were some differences in retention time and height among the peaks, and consequently, in their peptide profile. Therefore, there was a general similarity, but with individual differences.

Katsiari et al. [33] carried out HPLC analysis of water-soluble extracts of Feta cheeses made with NaCl or mixtures of NaCl and KCl during ripening and preservation. Moatsou et al. [39] investigated the nitrogenous fractions of traditional Feta cheese during ripening using HPLC analysis. Kocak et al. [40] evaluated peptide profiles in white-brined cheeses made from goat milk with adjunct cultures during ripening and preservation, using HPLC analysis. Sahingil et al. [41] evaluated peptide profiles of water-soluble extracts of white-brined cheeses during ripening and preservation using HPLC analysis.

In Table 5, the antioxidant activity and peptide level of the water-soluble extract of Feta cheese during ripening and storage is shown.

The water-soluble extracts of Feta cheeses have antioxidant activity. Perna et al. [24] determined the antioxidant activity of cow cheeses using the FRAP assay and it was found to increase during ripening and storage. Gupta et al. [19] showed that cow cheeses had significant peptide content using the Lowry assay, which was increased during ripening and storage. Gandhi et al. [42] determined peptide content using the Bradford assay in water-soluble extracts of brined cheeses. Reduction in peptide content of cheeses that were brined for 10 days was observed, and then peptide content varied depending on the brine composition.

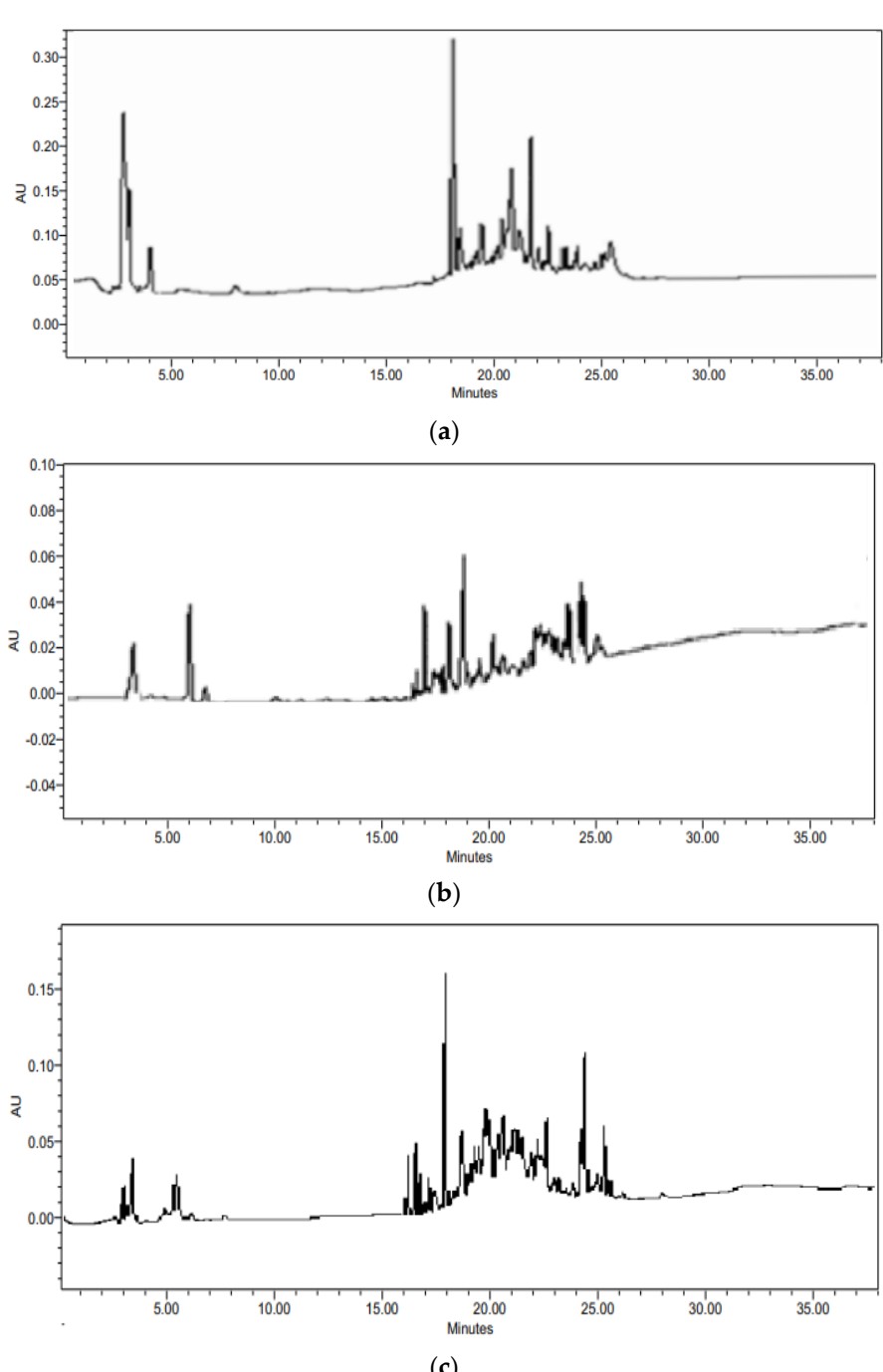

**Figure 1.** Typical chromatograms of Feta cheese (**a**), goat cheese(**b**) and cow cheese(**c**).

### 3.2. Metsovone Cheese and Other Ripened Smoked Cheeses

Table 6 presents the antioxidant activity and peptide content of the water-soluble extract of Metsovone cheese, aged 3 and 6 months, as well as other smoked cheeses aged 3 months from different regions in Greece.

The WSEs of Metsovone cheese and other smoked cheeses showed significant peptide levels. The values of the peptide content determined by the Lowry assay were higher in comparison to the values of the peptide content determined by the Bradford assay. The difference of peptide values among Bradford and Lowry assays were due to the different sensitivity of the assays [38]. The WSEs exhibited antioxidant activity, as determined by the Folin and FRAP assays. Shaibanl et al. [43] determined significant antioxidant activity in smoked cheeses using the Folin assay. The antioxidant activity was significant for

Metsovone cheese 6 aged months. Vosgan et al. [44] observed an increase of the antioxidant activity during the preservation of smoked cheese.

In Figure 2, the typical chromatogram of ripened smoked cheese is presented.

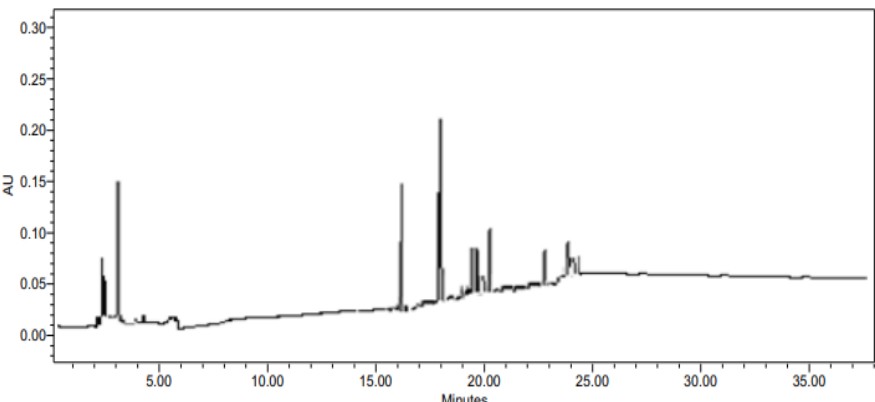

**Figure 2.** Typical chromatogram of smoked cheese.

In the first region of the chromatogram, two high peaks appeared. In the second region, a cluster of various peaks were observed. There was a general similarity with the chromatograms of various brined cheeses, while individual differences were also observed.

*3.3. Anti-Inflammatory Activity of Cheeses*

The anti-inflammatory activity of Feta cheese and other ripened brined cheeses, as well as Metsovone cheese and other smoked cheeses, is expressed as % inhibition. The anti-inflammatory activity of Feta cheese ranged between 5.95% and 8.73%. The brined goat cheese had a range of values between 3.25% and 7.67%. The values of anti-inflammatory activity of brined cow cheese ranged between 2.82% and 6.02%. The anti-inflammatory activity of Metsovone cheese and other smoked cheeses ranged between 8.27% and 12.2%. During the determination of anti-inflammatory activity, the mixture was not clear, so the water-soluble extracts of cheeses were diluted two times and the results had low values. The above findings suggest that peptides in the water-soluble extracts of cheeses may exhibit anti-inflammatory activity.

**4. Conclusions**

The water-soluble extracts of Feta cheese differ from brined goat cheese in both their antioxidant activity and peptide content. Furthermore, the water-soluble extracts of Feta cheese differ from brined cow cheese in their antioxidant activity. Feta cheese shows higher antioxidant activity compared to brined, cow and goat cheeses. Additionally, the peptide content of Feta cheese is higher than brined goat cheese. Metsovone cheese and other smoked cheeses show antioxidant activity and contain significant peptide levels. The water-soluble extracts of Feta cheese, as well as other brined cheeses, and Metsovone cheese, and other smoked cheeses, show some anti-inflammatory activity.

**Author Contributions:** Conceptualization, I.G.R.; methodology, I.G.R.; software, K.B.; validation, I.G.R., A.K. and I.L.; formal analysis, A.K. and I.L.; investigation, I.G.R.; resources, I.G.R.; data curation, A.K. and I.L.; writing—original draft preparation, A.K. and I.G.R.; writing—review and editing, I.G.R.; visualization, A.K. and K.B.; supervision, I.G.R.; project administration, I.G.R.; funding acquisition, I.G.R. All authors have read and agreed to the published version of the manuscript.

**Funding:** We acknowledge support of this work by the project "Development of research infrastructures for the design, production and promotion of the quality and safety characteristics of agri-food and bio-functional products" "(EV-AGRO-NUTRITION)" (MIS 5047235), which is implemented under the action "Reinforcement of the Research and Innovation Infrastructure", funded by the Operational Programme "Competitiveness, Entrepreneurship and Innovation" (NSRF 2014–2020) and co-financed by Greece and the European Union (European Regional Development Fund).

**Institutional Review Board Statement:** Not applicable.

**Informed Consent Statement:** Not applicable.

**Data Availability Statement:** Data are contained within the article.

**Acknowledgments:** We wanted to express our gratitude to the dairy industries, Dodoni, Karalis, Bizios, Erymanthos, Exarchos, Vlacha, Tositsa Foundation, Pappas and BelasVermion for the supply of cheese samples.

**Conflicts of Interest:** The authors declare no conflicts of interest.

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
