# Peer review of "Antioxidant Activity and Peptide Levels of Water-Soluble Extracts of Feta, Metsovone and Related Cheeses"

_applsci, doi:10.3390/app14010265_

Round 1

Reviewer 1 Report

Comments and Suggestions for Authors

Reviewer Comments

General comments:

The manuscript is well written. The authors investigated the “Properties of water soluble extracts of Feta, Metsovone and related cheese”. This study aims to evaluate the antioxidant activity and peptide levels of Feta cheese and other brined cheese, and Metsovone cheese and other smoked cheese. The study is interesting and adds to the existing body of knowledge. There are a few things that need clarification and revisions. The manuscript should undergo English editing/ proofreading. 

Details comments: 

1.     Page 1, Title of the manuscript: The title of the manuscript should be revised or changed because the study aims to determine the antioxidant and peptide levels of cheese. The word “properties” in the current title is too general and does not focus on the scope of the current study.

2.     Page 1, Lines 13-31 (Abstract): The abstract of the current study is too superficial. The authors should add a brief background of cheese and types of cheese. The methodology should be briefly explained. The anti-inflammatory assay should be added. The result mentions significant levels of antioxidant and peptide levels (increased or decreased? Please revise). The conclusion of the abstract should be revised.

3.      Page 1, Introduction: Please add the latest previous pharmacological properties of cheese.

4.     Page 2, Lines 47-50: Please briefly explain the mechanism of action of how cheese inhibits oxidative stress/neutralizes free radicals.

5.     Page 1, Lines 50-58: Please mention the parameters that have been measured in the previous study (to determine antioxidants).

6.     Page 2, Lines 65-67: Please rewrite the objective of the current study with the specific methodology used in the current study.

7.     Page 3, Lines 102-103: The cheese extracts are directly used after preparation of the store before the experiment? Please mention the storage condition of the extract.

8.     Page 3, Line 120: Perna et al. [20]

9.     Page 3 Line 132: Bovine serum albumin (BSA).

10.  Page 4, Lines 145-153: What equipment was used to determine HPLC analysis and software for analysis?

11.  Page 4, Line 165: Please rewrite the formula in separate lines or use the formula equation.

12.  Page 5, Tables 1-6: Please add the significant levels comparing each group of cheese if possible. How many replicates (n=?) for each cheese group?

13.  Page 6, Table 4: Please mention the color mark representation in the table footer. 

14.  Page 6, Line 223: Stobiecka et al. [14].

15.  Page 6, Line 228: significant peptide levels (Increase or decrease?)

16.  Figures 1-4: Please replace the figure with a high-resolution chromatogram. The current figure is unclear, and the peaks are not sharp.

17.  Page 12, Lines 374: Please rewrite the sentence that mentions that Metsovone cheese significantly increases or decreases the antioxidant activity and peptide levels.

18.  References: The author should revise the format of the references according to the MPDI reference format. 

Comments on the Quality of English Language

The manuscript should undergo english editing/prooreading.

Reviewer 2 Report

Comments and Suggestions for Authors

This manuscript presents an original study about the Properties of water soluble extracts of Feta, Metsovone and related cheeses.

The experimental results showed that Feta cheese and brined cow cheese, brined goat cheese, Metsovone cheese and other smoked cheeses exhibit antioxidant activity and significant peptide levels and some anti-inflammatory activity.

The technical quality of the manuscript is good in terms of how it was written and how the experimental results are presented. The style of expression reflects the scientific training of the authors. The manuscript is edited in accordance with the article drafting requirements.

The Abstract is concise and contains sufficient information to highlight the content of the article and the Introduction section provides a clear statement of the problem studied in the present manuscript.

The Materials and Methods section is well presented and appropriate to the purpose of the research.

Results follow the guidelines described in the Author's Guide and are good discussed.

References are relevant and current and follow the journal’s format.

The conclusions of the article fully reflect the results of the given study.

Please find below some suggestions/comments that might help in improving the quality of the manuscript:

Please check, in decimal numbers is used a point and not a comma.

Please use GAE when expressing in gallic acid.

L 115, 124, 133, 144: Add the calibration curve and the correlation coefficient.

L 251, 275, 292, 349: The chromatograms in figures 1, 2, 3 and 4 are not visible.

Please highlight peak regions corresponding to hydrophilic and hydrophobic peptides in the chromatograms.

L 158: What is LOX?

L 159: What is the concentration of ethanol?

Reviewer 3 Report

Comments and Suggestions for Authors

1.       Numerous language and grammatical errors make reading and interpretation of this manuscript difficult. Consider revising and refining the text extensively to enhance clarity, coherence, and overall readability.

2.       The purpose of the study in the manuscript was not clear and the need for the study was not demonstrated by simply comparing the antioxidant activity and peptide content of different cheeses.

3.       It is recommended that the authors should substantially revise the introduction to better reflect the innovation and significance of the study. The most important aspects related to this topic should be clearly presented in order to provide a properly description of the state of art in this field.

4.       Is the percentage concentration mentioned in the text weight ratio?

5.       Line 160: for 20 min?

6.       Line 165: The meaning of letters in the formula should be defined.

7.       Whether the indicators of different samples are significantly different from each other. (Table 1, 2, 3. 5 and 6)

8.       Section results and discussion is too descriptive and should be systematized. The experimental results included in each subsection should be clearly presented and discussed in accordance with the main objectives of this study.

9.       Since the main purpose of this paper is to evaluate the antioxidant properties, more antioxidant evaluation methods should be adopted. Specific methods can be found in the following literature.https://doi.org/10.3390/molecules28083384, https://doi.org/10.1021/jf1007665

10.     It might be better if the peptides in the samples could be identified.

Comments on the Quality of English Language

none

Reviewer 4 Report

Comments and Suggestions for Authors

Comments

1 Abstract lines 17-18, the ‘differ’ should be replaced by its past tense.

2 Lines 19-20 the expression ’cheeses exhibit antioxidant activity and significant peptide levels’ should be revised.

3 line 21 ‘suggesting that may peptides exhibit anti-inflammatory activity.’ Should be revised.

4 line 52 ‘while after fifth month a decrease’, the grammar errors should be corrected.

5 Line 65 ‘In the present study was carried out’ the word ‘in’ should be deleted.

6 line 87 ’ Water HPLC grade and acetonitrile HPLC grade’ should be revised.

7 line 110 ‘in the dark at room temperature for 2 h. After incubation in the dark at room temperature for 2 h’ these are the same meanings, only one sentence should be kept.

8 Line 113 the standard curve of gallic acid involving R2 value, and the linear regression range must be given.

9 Lines123-124, ‘’Gallic acid was used as a standard and the results were expressed as mg/L gallic acid equivalents. Moreover, were expressed as mg gallic acid/kg cheese.” How the authors use gallic acid as a standard to determine FRAP values? It is not suitable. The authors must use VE or VC as reference standard.

10 For the Folin-Ciocalteu assay, it can only be used to determine the amount of total phenolic compounds, How do the authors use it to determine antioxidant activity? As shown in Lines 106-114.

11 The authors stated that the peptide content was calculated using Bradford assay (lines 127-132 ), why the Lowry assay was used again to determine the peptide content?

12 The standard curve of BSA must be given for the determination of peptides.

13 How to determine the peptide only by HPLC, without any other information such as standard compounds? This method can not work.

14 For the ‘Statistical analysis’ part, it should be rewritten, it is too complex now.

15 For the tables 1 and 2, 3, the statistical difference should be marked for all data.

16 the resolution of all figures are too low. Furthermore, the authors must identify some primary peaks of all HPLC profiles, otherwise, it is no necessary to use these figures.

Round 2

Reviewer 1 Report

Comments and Suggestions for Authors

General Comments

The manuscript submitted to Applied Sciences with new title “Antioxidant activity and peptide levels of water soluble extracts of Feta, Metsovone and related cheeses” is significantly improved after the authors' corrections. The authors have appropriately responded to the original concerns. I think the changes are acceptable. I recommend the publication of the manuscript.  

Comments on the Quality of English Language

English language are acceptable. 

Author Response

Dear reviewer, thank you for your effort.

Reviewer 3 Report

Comments and Suggestions for Authors

It seems that the author is neither able nor willing to revise the manuscript as required. All comments for additional experiments were rejected without reasonable justification. It also seems that the author was unwilling to make it easier for the reviewers to see what changes the author had made to the manuscript, simply stating in his response to the reviewers that he had revised it without giving specifics about the changes.

Author Response

No comments

Reviewer 4 Report

Comments and Suggestions for Authors

Comments

1 the antioxidant activity cannot be determined by Folin assay.

2 Folin-Ciocalteu assay only used for the determination of TPC, total phenolic compounds. It is not for Antioxidant activity. It must be corrected and revised.

3 For the FRAP assay, the standard compound VC or VE should be used to calculate the antioxidant equivalent. Please refer to Antioxidants 2022, 11, 1543. https://doi.org/10.3390/antiox11081543

4 The Table 1, Folin must be TPC, the FRAP value must be expressed as VC or VE equivalent.

Author Response

No comments

Round 3

Reviewer 3 Report

Comments and Suggestions for Authors

It seems that the author is neither able nor willing to revise the manuscript as required. All comments for additional experiments were rejected without reasonable justification. It also seems that the author was unwilling to make it easier for the reviewers to see what changes the author had made to the manuscript, simply stating in his response to the reviewers that he had revised it without giving specifics about the changes.

Reviewer 4 Report

Comments and Suggestions for Authors

Accept in present form